# The Interactive Role of Family Functioning among BMI Status, Physical Activity, and High-Fat Food in Adolescents: Evidence from Shanghai, China

**DOI:** 10.3390/nu14194053

**Published:** 2022-09-29

**Authors:** Mingyue Chen, Wei Yin, Pauline Sung-Chan, Zhaoxin Wang, Jianwei Shi

**Affiliations:** 1School of Public Health, Shanghai Jiaotong University School of Medicine, Shanghai 200025, China; 2Department of Social and Behavioural Sciences, College of Liberal Arts and Social Sciences, City University of Hong Kong, Hong Kong, China; 3Hong Kong Institute of Economics & Business Strategy, HKU School of Business, The University of Hong Kong, Hong Kong, China; 4The First Affiliated Hospital of Hainan Medical University, Haikou 571199, China; 5School of Management, Hainan Medical University, Haikou 571199, China; 6Department of General Practice, Yangpu Hospital, Tongji University School of Medicine, Shanghai 200090, China; 7Department of Social Medicine and Health Management, School of Public Health, Shanghai Jiaotong University School of Medicine, Shanghai 200025, China

**Keywords:** BMI status, family function, physical activity, high-fat food, interaction, marginal effect

## Abstract

Objectives: Family functioning (FF), physical activity (PA), and high-fat food consumption (HF) are associated with adolescents being overweight and obese; however, little is known about their interactions. Therefore, this study aimed to examine how they work jointly on adolescent obesity with BMI as the outcome variable. Methods: A cross-sectional survey utilizing a cluster sampling design was conducted. Multinomial logistic regressions, multiplication interaction (MI), and marginal effects (MEs) were tested. Results: Active PA (non-overweight vs. obesity: OR = 2.260, 95% CI [1.318, 3.874]; overweight vs. obesity: OR = 2.096, 95% CI [1.167, 3.766]), healthy HF (non-overweight vs. obesity: OR = 2.048, 95% CI [1.105, 3.796]) and healthy FF (overweight vs. obesity: OR = 2.084, 95% CI [1.099, 3.952]) reduced obesity risk. Overweight students with healthy FF were less likely to become obese regardless of PA (inactive: OR = 2.181, 95% CI [1.114, 4.272]; active: OR = 3.870, 95% CI [1.719, 8.713]) or HF (unhealthy: OR = 4.615, 95% CI [1.049, 20.306]; healthy: OR = 5.116, 95% CI [1.352, 19.362]). The MEs of inactive PA and unhealthy FF were −0.071, 0.035, and 0.036 for non-overweight, overweight, and obese individuals, respectively (*p* < 0.05); the MEs of HF and healthy FF individuals were −0.267 and 0.198 for non-overweight and obese individuals, respectively (*p* < 0.05). Conclusions: Unhealthy FF regulated the influence of inactive PA or unhealthy HF on adolescent obesity, altogether leading to a higher risk of obesity.

## 1. Introduction

Being overweight or obese is defined as abnormal or excessive fat accumulation that may impair health. In 2016, over 340 million children and adolescents aged between 5 and 19 were overweight or obese [1]. Over the last fifty years, the obesity rate in adolescents has increased by approximately 5% per decade, and approximately one-quarter of children are now either overweight or obese [2]. Rapidly developing countries are showing a significant increase in childhood obesity [3]. In China, the overweight rate among children and adolescents aged between 6 and 17 was 9.6%, and the obesity rate was 6.4% in 2015, increasing from 2002 by 5.1% and 4.3%, respectively, demonstrating a rising tendency [4].

Obesity in children and adolescents harms both physical and mental health. Consequences mainly include: (1) continued obesity in adulthood; (2) increased risk of diseases, such as type 2 diabetes, cardiovascular disease, chronic kidney disease, and cancer; (3) increased death and premature mortality [3]. A growing body of literature links obesity to negative psycho-social consequences in adolescents, including poor emotional adjustment and stigmatisation, distorted perceptions of body image, antipathy from peers and adults, social withdrawal, and aggression, which are even more distinct in girls [5,6,7,8]. Currently, potential mediators of the association between being overweight and emotional well-being have become the focus. For example, one study showed that when body dissatisfaction was statistically controlled, the association between obesity and low self-esteem was no longer evident, nor was the relationship between obesity and depression [5,9]. These findings suggest that psycho-social factors should be taken into account when designing research or intervention programmes that deal with obesity [10,11,12,13].

In light of the above findings, interventions aiming to alleviate adolescent obesity should be considered. First, researchers have to identify the factors related to adolescent obesity. In general, physical activity (PA), dietary behaviour (DB), and family functioning (FF) are usually considered risk factors for adolescents being overweight and obese, despite several opposite conclusions. Physical activity has been a popular focus in this regard. It is well-established that regular PA benefits health and helps adolescents improve cardio-respiratory fitness, control weight, improve psychological well-being, and reduce the risk for chronic diseases [14,15]. Moreover, the relationship between PA and an epidemic of childhood obesity has been well-documented, in which PA is commonly regarded as a protective factor against adolescent obesity [16,17]. According to the Amsterdam Healthy Weight program, policies that encourage active transport and safe walking or cycling effectively reduce adolescent obesity [18,19].

The fundamental to reducing adolescent obesity is improving weight-related health behaviours, which basically balance energy intake and consumption [20]. Currently, the main perspective is still that high consumption of fried food or snacks high in sugar and fat leads to poor nutritional status and childhood obesity [21,22,23,24]. A meta-analysis concluded that sufficient consumption of fruits/vegetables, snacks, or fried food had a nonsignificant effect on childhood obesity, but consuming excessive sugar-sweetened beverages increased the risk of obesity by 24% [14]. According to the data from the China Health and Nutrition Survey (CHNS), the Chinese modern dietary pattern is characterised by high consumption of red meat and processed meat, as well as the use of the unhealthy cooking method of deep-frying, and such a dietary pattern abundant in saturated fat and cholesterol is reported to be positively associated with a higher risk of obesity [25]. Moreover, some studies have found that Chinese parents and grandparents are more inclined to push their children to eat a set amount of food at each meal, despite the child’s hunger and satiety signals, which causes common overeating among children and adolescents [26].

In addition to PA and dietary behaviours, environmental, genetic, biological, and socioenvironmental factors, especially family factors, are also crucial [3,13]. For example, research reported a positive correlation between household income and obesity, in which household income may alter the total calorie intake, DB, or PA of school children [27]. Among the studies on the interrelationship between family and children’s health behaviours, the prevailing conclusion is that parents influence their children directly through parental practices and role modelling, which affects children’s weight status [13,28,29]. In recent years, the family context has become an important factor in the formation of healthy behaviour in adolescents [13,28,29,30,31]. One of the key contextual factors of concern is family functioning (FF) [32]. In short, FF is how families cope with their day-to-day affairs by fulfilling their roles in the family [33]. Most studies showed a strong association between FF and adolescent obesity [31,34,35], but some studies concluded that there was no significant relationship between the two [36]. To our knowledge, no study has yet addressed the temporal association between FF and being overweight. FF was regarded as a moderator between health behaviours and parenting practices or parental modelling, but few existing studies have reached a consensus on its effects [29,37,38]. Multiple previous studies with small and accessible samples focusing on the relationship between FF and obesity-related behaviours have shown that healthier FF is associated with healthier dietary intake, less sedentary behaviour, and more physical activity, which is consistent with the finding of a national study among adolescents in the USA [29,31,33,39]. Children in dysfunctional families resort to effective but eventually dysfunctional eating habits, and they often consume energy-dense junk food to relieve emotional and stress-related pressure. All of these factors lead to poor self-rated health and severe obesity [8,40]. Therefore, a supportive and harmonious family environment is important for children’s growth and development [20,32,34].

In conclusion, PA, DB, and FF are all considered to be associated with adolescent obesity. Moreover, associations between FF, exercise, and diet have been well-documented in several studies, including those conducted among overweight adolescents [41]. Given these studies, health behaviours and FF may jointly influence childhood obesity and being overweight. However, few studies have examined the interaction role of PA, DB, and FF in exerting their influence on adolescent obesity and being overweight and, therefore, how they work together is still unclear [32,41].

This study provides updated and innovative evidence of the multiplication effects and marginal effects between FF and weight-related health behaviours, including physical activity and dietary behaviour, that occur daily on adolescent BMI so that more efficient early screening and prevention can be taken into practice to manage the global adolescent obesity epidemic.

## 2. Materials and Methods

### 2.1. Participants and Procedures

This study, conducted from September to December 2015, was a school-based cross-sectional health survey utilising a random cluster sampling design in Shanghai, China. The research team reached out to five junior high schools in Shanghai and introduced this study to them, with the aim of establishing collaborations. Among these, three schools in Jing’an District showed interest. All students in the 6th, 7th, and 8th (10 to 16 years old) grades of these schools were invited to take part in the study on a voluntary basis.

The sample size calculation principle was adopted in this study, which required that the sample size statistically cover ten times the number of items [42,43]. An 87-item questionnaire was synthesised to respond to the research question, with 9 items for sociodemographics, 3 items for physical activity, 15 items for dietary behaviours, and 60 items for family functioning. Hence, a sample size of 870 was the minimum. Considering a nonresponse rate of 20%, the target sample size was increased to 870/0.8 = 1088. The consent forms with descriptions of the study and the questionnaires were distributed to students and their parents by headteachers at the beginning of the semester. By returning the consent forms with both the parent’s and the student’s signatures, along with the completed questionnaires, the students indicated participation in the study. Out of 1088 students from three schools, 937 (86.12%) effective responses were received. A schematic of the study design is shown in Figure 1.

### 2.2. Measures

The data collection consisted of two parts: (1) anthropometric measurements conducted by trained researchers and (2) a self-report questionnaire completed by students that covered information about sociodemographics, physical activity (PA), dietary behaviours (DB), and family functioning (FF).

#### 2.2.1. Outcome Variables

Body weight (to the nearest 100 g) and height (to the nearest 0.5 cm) were obtained using a portable height and weight machine by the researchers. Respondents wore only light clothing and did not wear shoes during the measurement. For each student, two readings were taken for body weight and height, the averages of which were adopted for analysis. Body mass index (BMI) was computed following the formula BMI = kg/m^2^, where kg refers to the weight in kilograms, while m2 stands for height in metres squared. Age- and sex-specific BMI z scores were calculated based on a Shanghai local reference. BMI status was classified into three categories: non-overweight (underweight and normal weight) (BMI < 84th percentile), overweight (BMI ≥ 85th and <95th percentile), and obese (BMI ≥ 95th percentile) [44].

#### 2.2.2. Exposure Variables

Physical Activity (PA)

Selected items from the CDC Youth Risk Behaviour Survey were translated and reviewed by the research team experts to evaluate physical activity levels [45]. Indicators included days of exercise and the number of PE classes per week, which were rated on a scale from 1 to 7; in addition, participation in sports teams in the last year was rated from 1 to 4. The total score ranged from 3 to 9 after coding. A higher score indicated a higher level of physical activity. A score less than 4 was regarded as not active; otherwise, it was regarded as active.

#### 2.2.3. Dietary Behaviours (DB)

The questionnaire for dietary behaviours was constructed with reference to the Dietary Practices Questionnaire used in a qualitative study examining Hong Kong residents’ dietary practices [46]. There were 13 questions exploring dietary behaviours concerning (a) fruit and vegetable consumption (e.g., “What is the amount of your average daily intake of fruits?”), (b) protein and dairy consumption (e.g., “What is the amount of your daily intake of meat, fish, and egg?”), (c) high-fat food consumption (e.g., “What is the amount of your daily intake of fried food”), and (d) dietary habits (e.g., “Apart from meals, how often do you have snacks within a day?”). Respondents were asked to recall the frequency/amount of food consumption and their dietary habits for the past 7 days. A total score was calculated for each of the four categories. The total score ranges from 4 to 10 for fruit and vegetable consumption, 3 to 7 for protein and dairy consumption, 2 to 6 for high-fat food consumption, and 6 to 17 for dietary habits. A higher score indicates healthier behaviours. All scores were divided into two classes: unhealthy and healthy.

#### 2.2.4. Family Functioning (FF)

The McMaster Family Assessment Device (FAD) was employed to study family functioning [47] in our study. To our knowledge, the FAD is the only measurement that has a subscale that specifically assesses the general FF. There are seven subscales of the FAD: (1) problem solving (i.e., the ability of family members to solve problems to achieve good family functioning), (2) communication (i.e., the exchange of information between family members), (3) roles (i.e., if the family has regular patterns of behaviour to deal with family functioning), (4) affective responsiveness (i.e., the ability of family members to respond emotionally appropriately to environmental stimuli), (5) affective involvement (i.e., the degree of warmth among household members), (6) behavioural control (i.e., whether the family has a norm or standard to guide individuals’ reactions to emergencies), and (7) general functioning (i.e., the overall level of family functioning). Respondents were required to answer 60 items using a 4-point Likert scale ranging from 1 (strongly disagree) to 4 (strongly agree). Sample items are “We are able to make decisions about how to solve problems” (positive items) and “We cannot talk to each other about sadness we feel” (negative item); in addition, there are 35 positive questions and 25 negative questions. A mean score was computed by averaging the item scores after reverse scoring negatively worded items. For analytic purposes, the score was dichotomized. Participants were classified into the healthy FF group if their average scores were 3 or 4, while those with scores of 1 or 2 were classified as unhealthy FF [29,48].

#### 2.2.5. Adjusting Variables

Demographic variables were also collected, including sex, grade, age, length of stay in Shanghai, birth weight, primary caretaker, and monthly household income. According to the law on mandatory education in China [49], students of the 6th, 7th, and 8th percentiles were divided into three categories: ≤11, 12, and ≥13 years old. Length of stay in Shanghai was classified as ≤5, 5–10, and ≥10 years. Birth weight was grouped as ≤2.5 kg, 2.5–4 kg, and ≥4 kg [50]. Primary caretakers were classified into four groups: parents, grandparents, grandparents, and others. Monthly family income was divided into ≤CNY 3000, CNY 3001~6000, CNY 6001~9000, CNY 9001~12,000, and ≥ CNY 12,000. In addition, parental BMI was classified into non-overweight (underweight and normal), overweight, or obese (<24.0 kg/m^2^, 24.0–27.9 kg/m^2^, ≥ 28.0 kg/m^2^, respectively) [51].

### 2.3. Statistics

The data were analysed with Stata version 17.0. Descriptive data were calculated to indicate the prevalence of different BMI statuses. Frequencies and percentages were calculated for the remaining categorical variables. Chi-square tests were conducted to compare the difference in BMI status within the identified groups. Multinomial logistic regression analyses were used to calculate the odds ratio (OR) for “non-overweight”, “overweight” and “obese”. Multiplication interaction models were used to analyse the interaction of risk factors, including PA, HF, and FF, on BMI status. We then computed marginal effects at means following the logistic regression models. Statistical significance was defined as *p* < 0.05 (two-tailed).

## 3. Results

### 3.1. Sample Characteristics and BMI Status

Among the 937 participants (54.6% male), 36.71% were overweight or obese, with a mean BMI of 23.20 for overweight participants and 28.16 for obese participants (Table 1). The characteristics of the participants and the results of the chi-square tests are shown in Table 2. The average age of the sample was 12.47 ± 1.14 (M ± SD). The number of students from Grade 6, Grade 7, and Grade 8 was 362, 290, and 285, respectively. The average birth weight was 3.62 ± 1.04 kg, and 9% of students had macrosomia. Regarding the primary caretakers, 89.3% of students were cared for by parents, and others were usually raised by grandparents. In addition, approximately half of the students’ family incomes were more than CNY 9000 per month. A total of 53.7% and 24.1% of students’ fathers and mothers were overweight or obese, respectively. Among the sociodemographic variables, sex (*p* < 0.001), length of stay (*p* < 0.05), birth weight (*p* < 0.05), monthly family income (*p* < 0.05), and father’s BMI status (*p* < 0.001) were found to be significantly related to student BMI status (Table 2). In addition, the results showed that 27% of participants were physically active. There was a significant relationship (*p* < 0.05) between students’ physical activity and BMI status. Regarding dietary behaviours, only high-fat food consumption was related to BMI status (*p* < 0.05), with 9% of the students reporting having an excessive amount of HF. In addition, BMI status was found to be significantly dependent on family functioning (*p* < 0.05). Most of the students’ families (87%) had healthy family functioning.

### 3.2. Multinomial Logistic Regression Analysis of BMI Status

Multinomial logistic regression analysis was conducted to identify predictors of BMI status. In this model, the dependent variable was BMI status; variables that were found to be related to BMI status based on chi-square tests were entered as independent variables. The results are displayed in Table 3. Male students were more likely to be obese vs non-overweight (OR = 0.229, 95% CI [0.142, 0.369]) and overweight (OR = 0.538, 95% CI [0.316, 0.915]) than female students. The odds of having the BMI status of non-overweight (OR = 2.503, 95% CI [1.149, 5.542]) and overweight (OR = 2.672, 95% CI [1.075, 6.641]) were both twice as high as obesity in students whose birth weight was between 2.5 and 4.0 kg relative to those who were heavier than 4.0 kg. Students whose family monthly income was between CNY 3001 and CNY 6000 compared to more than CNY 12,000 had 2.4 times higher odds of being non-overweight than overweight (OR = 2.432, 95% CI [1.289, 4.589]). Students with non-overweight fathers had 2.8 higher odds of being non-overweight than obese in contrast to students with obese fathers (OR = 2.873, 95% CI [1.496, 5.517]).

Physically active students had higher rates of being non-overweight than overweight (OR = 2.260, 95% CI [1.318, 3.874]), as well as better odds of being overweight than obese (OR = 2.096, 95% CI [1.167, 3.766]) in comparison with physically inactive students, indicating that a healthy level of PA was a protective factor against adolescent obesity. Students who did not consume excess HF were less likely to become non-overweight as opposed to obese (OR = 2.048, 95% CI [1.105, 3.796]). Healthy FF reduced the odds of being obese compared to overweight compared to unhealthy family functioning (OR = 2.084, 95% CI [1.099, 3.952]).

### 3.3. The Multiplicative Interaction among PA, HF, and FF

To investigate how PA, HF, and FF work jointly to influence BMI status, we examined the multiplicative interaction among the three indicated variables. As shown in Table 4, there was a multiplicative interaction between FF and PA. Comparing the BMI status of non-overweight and obesity, students with healthy FF, whether their PA was active or not, had a lower risk of obesity, and those whose PA was active were more likely to stay as non-overweight (inactive: OR = 1.794, 95% CI [1.062, 3.030]; active: OR = 3.44, 95% CI [1.760, 6.722]). Overweight students who lived in families with healthy FF were less likely to become obese regardless of whether they exercised (inactive: OR = 2.181, 95% CI [1.114, 4.272]; active: OR = 3.870, 95% CI [1.719, 8.713]).

A similar trend was observed between FF and HF. Adolescents who maintained healthy high-fat food consumption in families with healthy FF were less likely to be obese (OR = 4.989, 95% CI [1.930, 12.899] for non-overweight). Those with healthy FF were at less risk of obesity even though they consumed superfluous HF (unhealthy: OR = 4.615, 95% CI [1.049, 20.306]; healthy: OR = 5.116, 95% CI [1.352, 19.362]).

### 3.4. The Marginal Effects between FF and PA or HF

Based on logistic regression, we tested the marginal effects of FF, PA, and HF on BMI status. As shown in Figure 2a, the MEs of inactive PA and unhealthy FF were −0.071, 0.035, and 0.036 for non-overweight, overweight, and obese individuals, respectively (*p* < 0.05). Figure 2b shows that the ME of unhealthy HF and healthy FF was −0.267 to non-overweight and 0.198 to obese (*p* < 0.05).

## 4. Discussion

The study set out to examine the multiplicative interaction among physical activity, high-fat food consumption, and family function with respect to their connections with adolescents’ BMI status. In this study, 36.71% of the students were overweight or obese, with a mean BMI of 23.20 kg/m^2^ for overweight and 28.16 kg/m^2^ for obese, which was much higher than the prevalence of being overweight and obese among children aged from 7–18 in China of 19.4% in 2014 (41,608/214,354) [52]. This may be because the study was conducted in Shanghai, which is a well-developed urban city. It is well documented that obesity rates among urban adolescents are higher than those in rural areas, which is associated with nutritional intake, physical activity, and air pollution due to traffic [53].

Connections between BMI status and several sociodemographic factors were identified in this study. First, consistent with a national investigation from China [52], boy students were more likely to be obese than girls. Possible explanations for this result concerns both children’s and parents’ health perceptions and behaviours. For example, Chinese parents tend to believe that physical development is different between boys and girls. This may lead to variance in feeding behaviours, mainly in the form of overfeeding boys, resulting in higher rates of boys being overweight and obese [54]. In addition, girls and boys have different perceptions of weight and diet, with girls feeling more pressure to be thinner and having lower confidence regarding their weight [53,54,55]. It also found that students with relatively low household incomes were more likely to be obese than non-overweight students compared to those with high monthly incomes. This finding was consistent with previous studies that revealed a negative association between income and obesity in China [56,57], and the burden of disease may shift to those of lower socioeconomic status in the future [56,57]. Likewise, studies from Europe and the USA also reported that decreasing family income and socioeconomic indicators were related to increasing the risk of childhood obesity and being overweight [58]; however, in low- and middle-income countries, affluence is positively associated with obesity [59]. Dietary behaviours could be a possible explanation for the difference. The high rate of being overweight among adolescents from economically affluent households may be due to a significantly higher weekly intake of meat or fish, eggs, dairy products, legumes, fruits, and vegetables than those from lower-income populations [56]. The high rate of being overweight in comparatively low-income households could be a characteristic of the disease burden of obesity in comparatively high-income countries. This is mainly because children from low-income families are more likely to eat unhealthy foods and develop unhealthy eating habits due to budget constraints [60]. Finally, the overweight and obesity rates in this study were associated with birth weight and paternal BMI, genetics, and nutrition during pregnancy, which may account for these connections [39,61,62].

The results concluded that unhealthy FF, inactive PA, and unhealthy HF were all risk factors for obesity. This is consistent with the findings of several previous studies [14,16,23,24,34,39]. Furthermore, we confirm our hypothesis that unhealthy FF and poor health behaviours have a synergistic influence on obesity in adolescents, together with increasing the risk to a large extent. There were MEs of inactive PA on being overweight and obese when family functioning was unhealthy and an ME of excessive HF when family functioning was healthy. Our findings are complemented by interaction effects that confirm the dynamic relationships among PA, HF, FF, and obesity. In particular, our conclusions add to the extensive literature on the influence of FF on children’s health behaviour, including those in obese populations [28,29,31,33,37,39,41]. A Canadian study demonstrated that FF was not an intermediate factor between parental practices and role modelling and healthy behaviour and, thus, it should be studied separately with regard to health behaviours [41]. However, similar to this study, a few studies are revealing that FF interacts with health behaviours. For example, it was shown that unhealthy FF was associated with less parental monitoring of their children’s diet and exercise [39]. Families with more conflict tend to pacify their children with tasty but high-fat food [63]. Moreover, previous evidence from intervention trials suggests that improved FF precedes weight-related changes, and that unhealthy FF has been shown to contribute to obesity [41,64]. These findings have shed light on designing obesity intervention programmes, where not only healthy behaviours but also FF should be included.

In summary, previous evidence suggested that DB, PA, and FF were all independently associated with BMI status, which was in line with our findings. Furthermore, our study is the first to prove that health behaviours and FF jointly influence adolescent obesity. However, the temporal order is still unclear [32,41]. With known findings, adolescents with health-related behaviour problems and obese students may be at greater risk of a poor family environment. Therefore, we suggest that clinicians assess the FF of these adolescents’ families. Both society and schools should seek opportunities to intervene in interfamilial problems [34,64]. Additionally, family members are expected to be aware of the importance of FF and strive to enhance family relationships through efficient ways, such as family health conversations [63,64,65]. Because this study is the first of its kind to investigate the interaction relationships among the indicated factors, one of the future directions could be replicating this study in more locations to gain sufficient data to verify the found associations. Moreover, cohort intervention studies that target improving family functioning in families with overweight/obese adolescents should be conducted to further clarify the temporal association between FF and health behaviours. Currently, there is a paucity of literature on how the seven subscales of FF influence specific behaviours [39,66]. Future research may be interested in linking FF and healthy behaviours in a more detailed manner to provide more precise directions for interventions.

The limitations were as follows: (1) the study adopted a cross-sectional design, so we could not establish causal or temporal relationships between FF and adolescent BMI and health behaviours. Additionally, there might be residual and unmeasured confounding. However, our adjustment for potential demographic confounders had no substantial impact on our association estimates. (2) The study was based on an economically developed urban city, where the economic level was higher than the national general average. Therefore, replications of this research in different socioeconomic backgrounds are recommended for future generalization. (3) Only fried food and desserts were included as high-fat foods. Future research may benefit from including more types of high-fat food, such as takeaway food.

## 5. Conclusions

Unhealthy HF, inactive PA, and unhealthy FF will increase the risk of being overweight and of obesity among adolescents. Unhealthy FF is a regulatory variable of either inactive PA or unhealthy HF increasing the risk of obesity in youths, which leads to a much higher risk, and healthy FF is also a regulatory variable of active PA or healthy HF helping maintain being non-overweight, which intensifies the effect.

## Figures and Tables

**Figure 1 nutrients-14-04053-f001:**
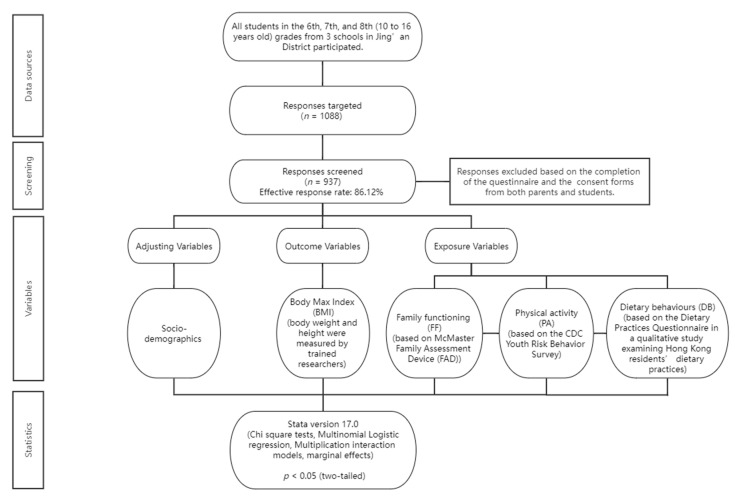
Schematic graph of the study design.

**Figure 2 nutrients-14-04053-f002:**
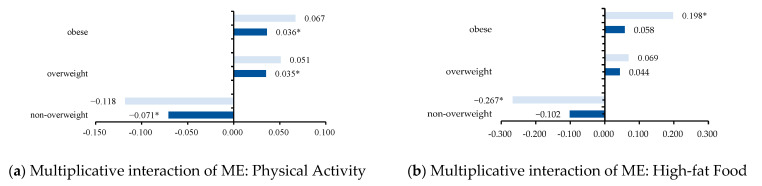
(**a**) Shows the marginal effect of PA at a specific value (PA = 1, which means inactive PA); the darker bars represent the marginal effect of inactive PA and unhealthy FF on BMI status, including non-overweight, overweight, and obese; the lighter bars represent the effect of inactive PA and healthy FF on BMI status; (**b**) shows the marginal effect of HF at a specific value (HF = 1, which means unhealthy intake of HF); the darker bars represent the marginal effect of unhealthy HF and unhealthy FF on BMI status; the lighter bars represent the effect of unhealthy HF and healthy FF on BMI status. * *p* < 0.05.

**Table 1 nutrients-14-04053-t001:** Descriptive statistics of BMI status.

BMI Status	*n*	%	*Mean*	*SD*	Minimum	Maximum
Non-overweight	593	63.29	18.27	1.98	13.39	22.75
Overweight	207	22.09	23.20	1.49	19.17	27.17
Obese	137	14.62	28.16	2.91	23.97	42.64

**Table 2 nutrients-14-04053-t002:** Sociodemographic, FF, PA, and DB of the sample according to BMI status.

Characteristics	Total		Non-Overweight		Overweight		Obesity		*X2*	*p*
	*n*	%	*n*	%	*n*	%	*n*	%		
Sex									56.552	<0.001
Male	512	54.64	272	45.87	133	64.25	107	78.10		
Female	425	45.36	321	54.13	74	35.75	30	21.90		
Grade										
6th	362	38.63	231	38.95	77	37.20	54	39.42	1.494	0.828
7th	290	30.95	187	31.53	60	20.69	43	31.39		
8th	285	30.42	175	29.51	70	33.82	40	29.20		
Age (years)										
≤11	215	22.95	132	22.26	52	25.12	31	22.63	0.748	0.945
12	279	29.78	179	30.19	59	28.50	41	29.93		
≥13	443	47.28	282	47.55	96	46.38	65	47.45		
Length of stay (years)										
≤5	50	5.34	33	5.56	13	6.28	4	2.92	11.014	0.026
5–10	779	83.14	478	80.61	177	85.51	124	90.51		
≥10	108	11.53	82	13.83	17	8.21	9	6.57		
Birth weight (kg)										
≤2.5	22	2.35	20	3.38	0	0.00	2	1.46	11.239	0.024
2.5–4	863	92.10	544	91.74	196	94.69	123	89.78		
≥4	52	5.55	29	4.89	11	5.31	12	8.76		
Primary caretakers										
Parents	799	89.27	514	90.18	178	88.56	107	86.29	7.904	0.245
Parental grandparents	50	5.59	30	5.26	9	4.48	11	8.87		
Maternal grandparents	39	4.436	20	3.51	13	6.47	6	4.84		
Others	7	0.78	6	1.05	1	0.50	0	0.0		
Monthly family income (* CNY)										
≤3000	49	5.63	27	4.84	12	6.25	10	8.26	19.204	0.014
3001~6000	187	21.47	140	25.09	31	16.15	16	13.22		
6001~9000	197	22.62	120	21.51	54	28.13	23	19.01		
9001~12,000	134	15.38	88	15.77	26	13.54	20	16.53		
≥12,000	304	34.90	183	32.8	69	35.9	52	43.0		
Father’s BMI status										
Non-overweight	434	46.32	303	51.10	89	43.00	42	30.66	20.727	<0.001
Overweight	402	42.90	234	39.46	95	45.89	73	53.28		
Obese	101	10.78	56	9.44	23	11.11	22	16.06		
Mother’s BMI status										
Non-overweight	711	75.88	463	78.08	149	71.98	99	72.26	6.335	0.175
Overweight	167	17.82	92	55.09	44	26.35	31	18.56		
Obese	59	6.30	38	6.41	14	6.76	7	5.11		
Physical activity										
Active	253	27.00	172	29.01	58	28.02	23	16.79	8.567	0.014
Not/less active	684	73.00	421	70.99	149	71.98	114	83.21		
Fruit and vegetable										
Unhealthy	208	77.80	453	76.39	166	80.19	110	80.29	1.861	0.394
Healthy	729	22.20	140	23.61	41	19.81	27	19.71		
Protein and dairy										
Unhealthy	276	29.46	172	29.01	65	31.40	39	28.47	0.499	0.779
Healthy	661	70.54	421	70.99	142	68.60	98	71.53		
High-fat food consumption										
Unhealthy	93	9.93	48	8.09	23	11.11	22	16.06	8.313	0.016
Healthy	844	90.07	545	91.91	31	88.89	26	83.94		
Dietary habits										
Unhealthy	184	19.64	118	19.90	39	18.84	27	19.71	0.109	0.947
healthy	753	80.36	475	80.10	168	81.16	110	80.29		
Family functioning										
Healthy	803	85.70	513	86.51	182	87.92	108	78.83	6.424	0.040
Unhealthy	134	14.30	80	13.49	25	12.08	29	21.17		

The primary caretakers and monthly household income index of 42 and 66 people were unknown, respectively. * Chinese yuan (currency unit). The average exchange rate between USD and the CNY in 2016 was 6.642.

**Table 3 nutrients-14-04053-t003:** Multinomial logistic regression analysis of BMI status.

	Non-Overweight (Reference: Obesity)				Overweight (Reference: Obesity)			
	OR	95%CI		*p*	OR	95%CI		*p*
		Lower	Upper			Lower	Upper	
Sex								
Male	0.229	0.142	0.369	<0.001	0.538	0.316	0.915	0.022
Female	1	reference		-	1	reference		-
Length of stay (years)								
≤5	0.722	0.195	2.667	0.625	1.661	0.400	6.900	0.485
5–10	0.535	0.248	1.150	0.109	0.997	0.414	2.401	0.995
≥10	1	reference		-	1	reference		-
Birth weight (kg)								
≤2.5	3.755	0.690	20.417	0.126	<0.001	<0.001	<0.001	0.981
2.5–4	2.503	1.149	5.452	0.021	2.672	1.075	6.641	0.034
≥4	1	reference		-	1	reference		-
Monthly family income (* CNY)								
≤3000	0.802	0.347	1.855	0.607	0.870	0.339	2.234	0.772
3001~6000	2.432	1.289	4.589	0.006	1.324	0.645	2.721	0.444
6001~9000	1.402	0.786	2.504	0.253	1.582	0.844	2.965	0.152
9001~12,000	1.509	0.811	2.809	0.194	1.007	0.495	2.047	0.985
≥12,000	1	reference		-	1	reference		-
Father’s BMI status								
Non-overweight	2.873	1.496	5.517	0.002	1.817	0.875	3.773	0.485
Overweight	1.381	0.729	2.617	0.322	1.407	0.689	2.873	0.995
Obese	1	reference		-	1	reference		-
Physical Activity								
Active	2.260	1.318	3.874	0.003	2.096	1.167	3.766	0.013
Inactive	1	reference		-	1	reference		-
High-fat food consumption								
Healthy	2.048	1.105	3.796	0.023	1.476	0.748	2.909	0.261
Unhealthy	1	reference		-	1	reference		-
Family Functioning								
Healthy	1.336	0.785	2.272	0.285	2.084	1.099	3.952	0.025
Unhealthy	1	reference		-	1	reference		-

* The independent variables were those with significant effects by the chi-square test.

**Table 4 nutrients-14-04053-t004:** The logistic regression of multiplicative interaction.

Multiplicative Interaction		Non-Overweight (Reference: Obesity)			Overweight (Reference: Obesity)		
Family Function	Physical Activity	OR	95%CI	*p*	OR	95%CI	*p*
Healthy	Active	3.440	(1.760, 6.722)	<0.001	3.870	(1.719, 8.713)	0.001
Healthy	Inactive	1.794	(1.062, 3.030)	0.029	2.181	(1.114, 4.272)	0.023
Unhealthy	Active	2.522	(0.790, 8.055)	0.118	2.941	(0.763, 11.336)	0.117
Unhealthy	Inactive	1		-	1		-
Family function	High-fat food consumption						
Healthy	Healthy	4.989	(1.930, 12.899)	0.001	5.116	(1.352, 19.362)	0.016
Healthy	Unhealthy	3.000	(0.982, 9.167)	0.054	4.615	(1.049, 20.306)	0.043
Unhealthy	Healthy	3.550	(1.244, 10.132)	0.018	3.300	(0.782, 13.930)	0.104
Unhealthy	Unhealthy	1		-	1		-

## Data Availability

The original contributions presented in the study are included in the article. Further inquiries can be directed to the corresponding authors.

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
