# Peer review of "The Interactive Role of Family Functioning among BMI Status, Physical Activity, and High-Fat Food in Adolescents: Evidence from Shanghai, China"

_nutrients, 2022, doi:10.3390/nu14194053_

Round 1

Reviewer 1 Report

This manuscript by Chen et al concluded that unhealthy family functioning regulated the influence of inactive physical activity or unhealthy high-fat food consumption on adolescent obesity, altogether leading to higher risk of obesity. Although the present study may provide some interesting information in the research field, below are some concerns I have for this manuscript. 

1.      In introduction section, author should first give brief background about obesity and its relationship, family functioning and physical activity.

2.      I suggest authors to make a schematic model which will help to understand the design of the study in a reader friendly mode.

3.      Please add relevant references wherever required, especially methodology.

4.      There are few grammatical and spelling mistakes which needs to be addressed.

Author Response

Thanks a lot!

Reviewer 2 Report

The manuscript "The Interaction Role of Family Functioning between BMI status, Physical Activity and High-Fat Food in Adolescents: Evidence from Shanghai, China" is a well designed epidemiology study. Conclusion is solid based on the data presented. However, the tone could be tuning down. For example, in line 25, consider replace "regulated" to altered or affected.

Author Response

We greatly appreciate your insightful comments, which we have used to improve the manuscript. Based on your comments and suggestions regarding the article’s language, we have asked the professional language company to revise the manuscript thoroughly. The changes are highlighted in yellow in the revised paper. We hope that the revisions render the manuscript more acceptable and read-friendly.

Thanks again!

Round 2

Reviewer 1 Report

Author has responded to all the queries.